# Teaching Social Marketing Using E-Service Learning Amidst Health and Humanitarian Crises: A Case Study from Lebanon

**DOI:** 10.3390/ijerph191912696

**Published:** 2022-10-04

**Authors:** Marco Bardus, Khawla Nasser AlDeen, Tamar Kabakian-Khasholian, Mayada Kanj, Aline Germani

**Affiliations:** 1Institute of Applied Health Research, College of Medical and Dental Sciences, University of Birmingham, Birmingham B15 2TT, UK; 2Department of Health Promotion and Community Health, Faculty of Health Sciences, American University of Beirut, Beirut 1107 2020, Lebanon; 3Center for Public Health Practice, Faculty of Health Sciences, American University of Beirut, Beirut 1107 2020, Lebanon

**Keywords:** social marketing, public health, universities, blended learning, service learning

## Abstract

We present the design, implementation, and evaluation of an e-service learning course, “Social Marketing for Health Promotion”, offered to full-time and part-time students enrolled in the Master of Public Health at our institution. In a quasi-experimental trial, we introduced e-service learning in 2018, comparing a traditional face-to-face section to a blended course (33% online). Based on the positive feedback received, we progressively increased the online component in the following academic years, reaching 100% online in Fall 2020. We compared the quantitative and qualitative indicators evaluating three e-service learning-course iterations with a face-to-face control. The impact indicators included participation and engagement in the course, the attainment of the learning outcomes, satisfaction with the course, instructors and mode of delivery, and the impact of the experience beyond the classroom. Over the years, we trained 73 students whose engagement with the course remained relatively stable. The attainment of the learning outcomes and general course satisfaction steadily increased over time, demonstrating a positive impact on student learning. Qualitative data illustrate the importance of instructors in setting expectations and guiding students and community partners through a remote-learning process.

## 1. Introduction

Public health is a field that is grounded in the values of justice, social change, and community service, which makes it the perfect discipline to adopt service learning. Service learning is an experiential-learning pedagogical approach that blends academic learning with service to the community [1]. This approach has become commonly used to build sustainable relationships between universities and local communities. Service learning has also been instrumental in creating a new public health workforce capable of empowering and developing healthy communities [2,3,4]. Utilizing service learning in public health has mutual benefits for the students, professors, institutions, and community partners [2], showing positive effects on student learning outcomes [5,6]. By developing collaborative relationships between educational institutions and communities, experiential learning provides all the participating agents with an opportunity to apply community-driven approaches in service while practicing collective and intellectual civic engagement [4]. Through this reflective practice in learning, public health students can practice intentional volunteerism and apply the lessons learned from the curriculum at the real-time grassroots level [2,4]. This experience allows students to achieve their course learning objectives and public health competencies dynamically and interactively while providing community partners with active volunteerism to meet operational needs and build capacity [2]. The authentic partnerships stemming from service learning serve as a bridge between academia and communities through the professors and students, allowing the translation of the latest evidence to field service; for the community, service learning in public health can promote health equity and can mindfully cater to the community needs [3,4]. Examples of service learning applications in public health education include biostatistics courses [7], social and behavioral principles of public health [8], nursing [4], healthcare management [9], or general public health [2,3].

Based on recent publications that reflect on our experiences of promoting sustainable behaviors through service learning activities [10,11], this case study elaborates on our multiyear experience of running a social marketing course at the American University of Beirut before, during, and after the introduction of an online learning component.

## 2. Literature Review

### 2.1. Social Marketing Education and Service Learning

Social marketing has gained notoriety and fortune in the public health community worldwide [12]. Developed as a discipline and practice in the 1960s and 1970s (see the seminal paper by Kotler and Zaltman [13]), in the last 30 years, social marketing has emerged as a distinct field of study that combines multidisciplinary approaches with social and behavior change [14] that is applied to health promotion [15,16,17] and environmental health [18,19]. It is a recognized planning approach endorsed by the European Centre for Disease Control (ECDC) [20], among other agencies. The global consensus definition, endorsed by the professional associations?, recites that social marketing “seeks to develop and integrate marketing concepts with other approaches to influence behaviors that benefit individuals and communities for the greater social good” [21]. Social marketing fosters the strategic programmatic thinking necessary for designing behavior-change programs and initiatives [17].

In 2014, the Australian and European social marketing associations promoted and endorsed educational standards that could serve as benchmarks for academic course development and professional accreditation [22]. Not surprisingly, the Association of Schools of Public Health in the European Region (ASPHER) recently listed social marketing under the “intellectual competencies” of public health professionals (Chapter E.1), and among the “central concepts applied to health promotion” (E.1.4). Social marketing is also linked to “basic theories underlying communication skills “(E.1.6) [23]. Similarly, the U.S.-based Association of Schools and Programs of Public Health (ASPPH) mentions marketing under the competencies for Health Policy and Management (“applying principles of strategic planning and marketing to public health”) [24]. Even though social marketing programs are few [25] and they are taught chiefly in business and communications schools [26,27,28], they are offered mostly in the “Global North” (North America, Europe, and Australia) [29], some courses are offered in schools of public health, with some cases of its application reported in the Middle East, Latin America, and Africa [29].

In Europe, full academic courses in social marketing are taught within graduate degree programs in health communication or marketing, offered by various universities in the United Kingdom (e.g., Brighton, Nottingham, Dublin City, Ulster, or Huddersfield), Slovenia (the University of Ljubljana, Faculty of Social Sciences) [30], and Switzerland (the University of Lugano, Faculty of Communication Sciences), where the local School of Public Health offered a special edition of the 2019 Summer School focused on social marketing [31]. The relatively slow uptake of social marketing in public health schools might be due to the competition with other disciplines (e.g., behavioral economics) [30] or to the limited capacity of public health faculties to teach marketing concepts in an innovative way [28]. Social marketing courses are taught face-to-face, using traditional pedagogical approaches, such as frontal lectures and discussions. Social marketing courses generally culminate in the development of a social marketing plan [26,27], which is an activity that aims to apply social marketing principles to solving a “wicked problem”, be it fictitious, realistic, or actual [32]. Hence, marketing courses need to be more relevant, helpful and appealing to public health students and faculties [28].

Because social marketing aims to foster healthy communities, service learning can be considered the ideal pedagogical approach for teaching a practice-oriented approach such as social marketing. According to Domegan and Bringle, service learning is particularly fitting for social marketing education because it instills civic engagement and social responsibility among students; at the same time, social marketing benefits service learning activities as it develops plans that can be applied to solve actual public health problems [33]. However, the literature on the application of service learning in social marketing education is limited; to the best of our knowledge, there are only a few examples of published cases that show the positive impact of this pedagogy on student engagement and community-academic partnerships [10,34], and satisfaction with the learning process [32].

### 2.2. E-Service Learning

Online and distance education has witnessed significant growth in recent years, but the COVID-19 pandemic has steered academia towards reimagining its role in providing essential educational services [35,36]. Furthermore, overstretched healthcare systems have relied globally on student volunteers’ efforts to meet the heightened needs, driving medical, allied health, and public health programs to rapidly transform their curricula and pivot toward a community-based response [4,37,38,39]. Hence, the COVID-19 pandemic has forced higher-education institutions to rethink service learning while following the public health guidelines for reducing the risk of viral transmission. Despite some scholars raising concerns about the limitations of computer-mediated learning [40], the COVID-19 pandemic has increased the adoption of “e-service learning” in various educational settings [41,42,43,44]. E-service learning can be defined as the application of service learning in which both the instructional and service components can be performed entirely or partially online [9]. Similar to traditional service learning, e-service learning requires a delicate balance between the knowledge acquired from the course materials and the time dedicated to applying this knowledge in projects catered to real-life scenarios [45,46]. Some recent evidence claims service learning can effectively catalyze knowledge transfer and community engagement while eliminating the traditional geographical boundaries imposed by face-to-face experiences [46].

The e-service learning models vary depending on the education field and service area, but they all benefit the stakeholders involved [47,48]. Several conceptual papers provide the rationale for the benefits of e-service learning for students, instructors, and higher-education institutions. For the students, e-service learning is similar to on-site experiences, enhancing soft critical skills, such as critical decision-making, time management, emotional intelligence, interpersonal and group communication, etc. [45,46,47,49]. For higher-education institutions, e-service learning can foster civic engagement, developing strong ties with local communities [47,50]. Some case reports have shown that an essential success component in the implementation of e-service learning courses was building the course around the needs and calendars of the community rather than around the educational calendar and course competencies [50]. Other essential factors include contextualizing the service learning component of a course [46], and the role of instructors as mediators or facilitators in the community–university learning process [45,46,47].

The evidence on blended learning’s effectiveness is somehow mixed and inconclusive. While some meta-analyses support the hypothesis that blended learning yields better learning outcomes [51,52], some primary research studies show the opposite or no results [53]. Moreover, some recent systematic reviews on e-service learning identified only a few empirical applications that can partially demonstrate the e-service learning-course effectiveness [46,54]. For example, Waldner and colleagues’ 2012 review identified 18 reports on courses, one of which was about a marketing course, presented in 2006 at a conference [54]. Similarly, the more recent review by Stefaniak identified 24 studies covering education, communications, and humanities, but no marketing, public health, or social marketing examples [46].

Recent exemplar applications of e-service learning in public health education during the COVID-19 pandemic include a project in northern Australia [38] in which allied health students served in organizations that provided services to community members living in remote areas had disabilities or who had experienced injuries. The project leveraged robust partnerships with aged-care facilities. The students created a safety protocol to continue essential in-person services when possible. In addition, they developed online capacity-building sessions for the health workers at the facilities, sharing evidence-based literature findings and online training. These collaborations with a local community organization allowed the allied health services to continue during the pandemic. They let the students meet their practice-based program requirements and quickly graduate [38].

Another relevant example is a COVID-19-centered course offered at the University of Alabama at Birmingham [41]. The course included students from different disciplines, and the curriculum constituted educational guest-lecturer modules and infographic design team projects. These interactive assignments were used in community health-promotion campaigns, creating a sense of social responsibility among students [41]. Nevertheless, little is known about the application of e-service learning in other contexts and countries, particularly those in the “Global South”, which some scholars define as “countries classed as low- or middle-income by financial institutions, including the World Bank”, as Cateriano et al. recently discussed [29]. Although the definition of the “Global South” is controversial and uneasy, according to Mahler [55], it is a critical concept that refers to the nonterritorial condition of spaces and people that are negatively affected by globalization, acknowledging that “there are Souths in the geographic North and Norths in the geographic South” ([56], page 32).

Lebanon certainly belongs to this group of spaces, especially after the most recent social, political, economic, and financial crises [57,58], which still affect the small country in the eastern Mediterranean region. A fiscal and financial crisis emerged in October 2019 [58], revealing deep-rooted sociopolitical issues. The COVID-19 pandemic, which started in February 2020, and the Beirut Blast in August 2020 [59], shook the pillars of an already weakened healthcare system, exacerbating the underlying social and political inequities. These circumstances made it difficult to prioritize food, health, education, and other essential services, such as water, electricity, and the Internet, which has recently become a commodity. In Lebanon, the national power grid is so limited and dysfunctional that a parallel decentralized system provides electricity to households and institutions through diesel-based generators [60]. Prone to corruption and monopoly, the availability of electricity and the Internet depends on the supply and demand of constantly increasing fuel costs. On some occasions, Internet providers did not have the fuel to power their generators, resulting in significant service disruptions, especially in rural areas outside the capital Beirut. This has likely affected the students’ learning experiences and made it difficult for us to implement online-based education, even though the university pushed it.

## 3. Aims and Objectives of the Study

This case study aims to provide a comprehensive reflection on our three-year experience using e-service learning in a social marketing course in the context of protracted sociopolitical, economic, health, and humanitarian crises. In this case study, we evaluate the overall students’ experience and the impact of e-service learning on the student learning outcomes and community-partner organizations served. This study contributes to the limited empirical evidence on e-service learning in social marketing.

## 4. Materials and Methods

This case study builds on recent reflections on the role of service learning in social marketing education [10,61], comparing quantitative and qualitative data from the implementation of four iterations of the course offered in 2017, 2018, 2019, and 2020. We compare the results of a traditional face-to-face (0% online) version of the course in 2017 to those obtained through three e-service learning iterations. We introduced e-service learning in 2018, following a hybrid type III e-service learning model, following the classification by Waldner and colleagues [54]. In 2018, we had 4 online sessions out of 12 (33% online); the 2019 and 2020 iterations of the course included 60% and 100% online service learning, respectively. The pilot e-service learning course in 2018 was supported by an intramural grant from the Center of Teaching and Learning (Scholarship for Teaching and Learning). We secured ethical approval from the Institutional Review Board for the research component of this project (reference number: FHS.MB.06/SBS-2017-0332). The subsequent iterations of the course were built on the feedback received during the pilot research study. In the following paragraphs, we present some information on the context in which the course took place. We describe a timeline for each course iteration and the instruments used to assess our objectives.

### 4.1. The Course and Context

“Social Marketing for Health Promotion (HPCH 333)” is a 14-week 2-credit course (approximately 4 ECTS) offered to Master of Public Health students concentrating on health promotion and community health and to public health nutrition. The course is part of the Graduate Public Health Program hosted at the Faculty of Health Sciences at the American University of Beirut (AUB), Lebanon. Founded in 1866, the AUB is one of the oldest universities in the country. It is a middle-sized private university ranked 220th in the QS world rankings and 2nd in the Arab world [62]. The Faculty of Health Sciences, founded in 1954, is considered a leading center in the Arab world for studying public health. Its mission is to prepare professionals in the scientific disciplines of public health. The New York State Education Department accredits all its degree programs. In addition, the FHS sought and obtained accreditation for its graduate programs from the U.S. Council on Education for Public Health (CEPH).

Since the academic year 2012–2013, the course has included a service learning component that is based on a problem- and discipline-based model [63], whereby students are expected to develop strategies based on the social marketing framework, acting as “consultants” for a “client” (i.e., a community-partner organization). The first author of this paper has been the primary course instructor since 2016; the third author taught the course between 2011 and 2015 and co-taught the pilot e-service learning iteration of the course in Spring 2018. Each semester, the instructors, with the support of the Center for Public Health Practice (CPHP) at the Faculty of Health Sciences, identify collaborating community-partner organizations to codevelop a social marketing plan for a specified issue identified with the community-partner organization. Community-partner organizations include local and international nongovernmental organizations, municipalities, or departments of the Ministry of Social Affairs and Public Health, and AUB departments or units. At the beginning of the course, students choose the organization and a problem to work on throughout the semester. They start interacting with the community-partner organization to develop the social marketing strategy.

Traditionally taught face to face through lectures and assignments, the service learning component of this course included several field visits to the community-partner organization, which were timed to specific learning outcomes throughout the semester, and it was linked to the development of the primary assignment: the social marketing plan. The other essential activities included a reflective essay, in which students were asked to provide detailed critical accounts of their learning experiences, following Boud’s typology of “reflection after events” [64]. For the course taught in Spring 2017, a student-led project promoting waste segregation at the source generated a case study [11] that was included in the 2017 edition of the book “Social Marketing: Rebels with a Cause” [65].

### 4.2. Development of E-Service Learning Course

#### 4.2.1. The First Blended Course (33% Online)

In the Spring 2018 semester, we introduced a hybrid e-service learning model through a Scholarship of Teaching and Learning grant; the content of the course was developed using a student-centered design approach based on formative focus groups with eight students who took the course in the previous academic year. A research assistant conducted the focus groups to minimize social-desirability bias. The first and third authors were the co-instructors of the course and undertook blended-learning training offered by the institutional Office of Information Technology.

A quasi-experimental design was used to test the effect of the intervention (i.e., blended learning) on engagement in the course, interaction with the instructors, satisfaction with the course, and knowledge about social marketing. As per the registration requirements, it was not possible to randomly allocate students, so those who enrolled in the course could choose between traditional and blended sections—after reading a summary of the two options. We had to follow the university policy for blended courses, which caps the proportion of online sessions at 40% of the course. Hence, the blended section included: 4 online lessons out of 12 (33% online), followed by critical forum discussions and 8 face-to-face sessions to recap the online work. The traditional section included 12 face-to-face lessons and class discussions. In both sections, the students were encouraged to use online collaborative writing software (Google Docs), and web-based project-management software (Basecamp) to organize their work, communicate with the instructors, and, optionally, with the community partners.

Of the 13 students enrolled, 8 chose the blended section, and 5 the traditional section. The students chose sections only because they suited their schedules better. The students in the blended section were divided into two groups: the first partnered with the Municipality of Sin El Fil to develop an intervention promoting physical activity in schools; the second partnered with Skoun, a nongovernmental organization focused on substance-misuse prevention. The students in the traditional section collaborated with SAID, a nongovernmental organization focused on colorectal-cancer prevention.

A summative focus-group discussion at the end of the Spring 2018 semester with ten students revealed that many in the blended section liked the whole practical experience of service learning and evaluated their experiences positively. Most of the students deemed the course appropriate and well-paced, as it helped them to rely on themselves and to have more time to work with the community. Many appreciated the flexibility of accessing the course at their convenience (“I like the blended course because I can access it whenever I find a good time for me to do so”). Some students lamented that the online lectures were “non-interactive enough” (as they were mostly voice-over PowerPoint recordings). Another limitation a student mentioned was the limited time to engage in online activities. The feedback received in the pilot intervention allowed the instructors to adjust the course content by introducing interactive Moodle lessons, feedback via live meetings on Zoom, and smaller-stake activities to reduce the perceived workload and allocate more time for the service activities.

#### 4.2.2. Hybrid Edition (60% Online)

Starting from October 2019, Lebanon, which the World Bank by then considered an upper middle-income country, was hit by compounded and unprecedented political and economic crises [66], resulting in the generalized “Lebanese Uprisings” [67]. The social and political turmoil forced the government to resign and the country to a halt through street revolts, road blockades, strikes, and riots. In the middle of the fall semester, the deteriorating security situation urged the AUB administration to suspend face-to-face classes—and when these were possible, some students could not reach the university due to roadblocks. This forced the instructors to increase the number of online lessons, reaching up to 7 out of 12 sessions (60% online). The instructors started creating voice-over PowerPoint recordings of missed lectures and delivered the rest of the weekly classes via Zoom, with some students in class and others at home. The instructors also posted session recordings on the YouTube channel to allow those unable to attend the face-to-face meetings to follow the discussions.

Given the difficulties in moving around the city due to the frequent roadblocks and revolts, community-partner organizations were sought within the AUB. The instructors scouted four pro-environmental issues linked to three units: the 20 students enrolled, divided into 4 groups out of 5, worked with: the AUB Botanic Committee to promote physical and mental wellbeing throughout the university campus; the Neighborhood Initiative to encourage the sorting of recyclables and cigarette-butt recycling in the community surrounding the campus; the Environmental Health Safety and Risk Management department to promote organic-waste composting among the AUB campus residents.

#### 4.2.3. Full-Fledged E-Service Learning (100% Online)

In 2020, the economic and financial crises deteriorated further, leading to a decline of 19% in the GDP and a triple-digit inflation [68]. In February 2020, COVID-19 hit the country, urging the existing caretaker government to authorize a series of nationwide lockdowns to curb the diffusion of the virus. These policy decisions further exacerbated the economic and political situation. On August 4, 2020, a large amount of ammonium nitrate stored haphazardly in the Beirut Port generated an explosion estimated to be between 407 and 936 tons of TNT [69]. The explosion annihilated a large part of the capital city, plunging the country into a deep humanitarian crisis, from which it has not yet recovered [57,59]. Considering the situation, the university administration was forced to close the campus and move to remote courses from the beginning of the fall semester. At the same time, the university could not cover the Internet costs for the students who needed to stay at home and whose families had to incur additional charges to provide a larger bandwidth for the online courses.

Consequently, the instructors designed a series of prerecorded lectures on Moodle, including YouTube video recordings introducing the class topics. The instructors produced seven self-paced lessons and introduced a new strategy to evaluate student participation and stimulate engagement, called the “Karma Participation” (KP) system. KP is a point-based system that awards students participation points for actively engaging in the classroom, completing tasks, participating in discussions synchronously or asynchronously, or evaluating others’ work throughout the course.

Like the previous semester, community-partner organizations were sought within the university, and the proposed projects focused on COVID-19-related issues. Students worked with the Faculty of Health Sciences COVID-19 Public Health Initiative to develop an intervention that provided mental health support for AUB students, with the Center for Public Health Practice to develop an intervention that promoted mask-wearing, physical distancing, and hand hygiene among Syrian refugees, and with the Humanitarian Engineering Initiative to create a platform to address COVID-19-related misinformation.

### 4.3. Impact Indicators

To address the research questions, we examined the impact of the course on students’ learning, such as (1) engagement and participation in the course, (2) the attainment of the learning outcomes, and (3) satisfaction with the course, instructors, and mode of delivery. Finally, we considered the impact on the community organizations served beyond the classroom.

**Engagement and participation in the course**: This was assessed through a course-participation grade worth 5% in 2017–2018, 10% in 2019, and 30% of the final grade in 2020. In Spring 2017 and 2018, the grade was based on the Moodle activity logs and the instructors’ subjective evaluations; in Fall 2019 and 2020, the participation grades included a combination of indicators, such as the attainment of self-paced lessons and online discussions, peer evaluation, community-partner evaluations, and the instructor’s evaluation. Due to these variabilities, the grades cannot be compared from a statistical standpoint.

**Attainment of learning outcomes**: This was assessed through the final-project grade (i.e., the social marketing plan) and the student-based instructor-course-evaluation (ICE) reports (Section C). The ICE reports are collected at the end of each semester by AUB’s Office of Institutional Research and Assessment (OIRA). They include quantitative and qualitative feedback from the students enrolled in the course. An abstract of the ICE-report data is contained in the appendix.

**Satisfaction with the course, instructors, and mode of delivery:** This was measured through ICE reports and included quantitative evaluations—ratings (Section A for the instructor, Section B for the course, Section D for the online experience)—and qualitative feedback. We also used quotes from the written reflections collected at the end of the semester that mentioned the role of technology and e-service learning.

**Impact of the course beyond the classroom**: This included evidence from students’ reflections and the implementation of the social marketing plans following the course.

## 5. Results

### 5.1. Participants and Topics Addressed

Figure 1 summarizes the four editions of the course with the number of students enrolled in each iteration, the topics covered, and the related community-partner organizations.

### 5.2. Impact of E-Service Learning on Students’ Outcomes

A summary of the impact indicators used in the course is included in Table 1. Overall, the participation grade in the pilot intervention in Spring 2018 was higher than the other iterations in 2019 and 2020, but it was like the participation grades recorded in 2017; the differences across the iterations were not statistically significant.

According to the final-year ICE reports, the attainment of the learning outcomes steadily increased from 3.8/5 in 2017 to 4.8/5 in 2020. The final-project grade varied significantly across the iterations: in Spring 2018, the final-project grade was higher in the traditional compared with the blended section, but in the other two iterations, the final-grade project was higher than the 2018 traditional section.

Throughout the four iterations of the course, the different cohorts of students showed increased levels of satisfaction with the course, instructors, and mode of delivery, reaching the maximum ratings in Fall 2020.

In Fall 2020, 10 out of 12 students completed all the self-paced lessons, and they spent, on average, 332 min on them; they completed 83% of the online activities. They also achieved 92% for the forum discussions. The final average participation grade was 83, which can be considered a good achievement for the first time the KP system was introduced to stimulate online and offline engagement.

### 5.3. Qualitative Feedback on E-Service Learning

The qualitative feedback provided by the students in the final ICE reports supported the overall positive quantitative ratings. For example, a student enrolled in the Spring 2018 blended section commented: “I liked that some of the class sessions were taught online, this has exposed us to a new interesting learning opportunity, and it was a great experience”.

In Fall 2019, a student, in the final reflection, elaborated on the role of the instructor in supporting the effective implementation of the service learning projects:

“It was unfortunate not to be able to meet because of geographical location differences. The pressure of wanting to meet but not being able to only increased our frustration and desire to reach the project’s goals. In the absence of classes, we learned to be self-sufficient and coordinate at a distance online with our professor and with each other to complete the project. Our professor cared about our safety, used Zoom online sessions, answered our questions promptly, and supported our right to protest when our country needed us”.

Another student that reflected on the learning experience, and the concurrent historical events that they were living in, looked at how the course became a testing ground for their future career:

“The *Thawra’* (revolution) has been a clear example of a real-life situation where [I had] to continue working on my project while adapting to a rapidly changing situation. I had to re-plan our working schedule from the beginning. Besides, we had to solve logistics with the target population and the strategies to reach them. As professionals, we had to find a way to agree very fast [on solutions] and re-think the best possible ways to continue to work efficiently and deliver quality work on time. Moreover, I believe that even the situation affected the way we were planning the intervention keeping it as cost-effective as possible, considering the current economic crisis. All the experiences we had are priceless; the obstacles we faced made us feel more serious about the project since it felt like a real-life thing that needs emergency solutions for future success, and these are precisely the skills that we need once we are out there, working as a public health professional”.

In Fall 2020, a comment in the ICE reports specifically mentioned the technological components that they appreciated the most:

“[The instructor] managed to turn this course into a fully online course with an interactive component: self–paced e-lessons and live sessions for discussions and activities as a team. He adapted the material for us very quickly and found creative ways to keep us engaged with the material”.

Several students appreciated the so-called “Karma Participation” point system, which made them feel more motivated, less isolated, and consequently more involved in the course. Other benefits of e-service learning were related to the perceived accessibility of the materials through the learning management system. Students could watch and re-watch the self-paced lessons, gaining flexibility and the ability to learn at their own pace. Many students also appreciated the virtual live meetings, which allowed them to discuss with the rest of the class and clarify unclear topics.

Despite many perceived benefits of e-service learning, some students lamented some of the medium’s limitations. In Spring 2018, a student stated in the focus group: “(I am) used to interacting with the instructors face-to-face, so I felt overwhelmed with making time for online classes”. A similar consideration came from a student enrolled in Fall 2020 who mentioned that e-service learning might not be appealing to all types of students if they are shy or not ready to engage with members of the public remotely:

“One important issue to reflect on is how challenging online communication was. Unfortunately, it was hard for me to engage a lot with the team, especially since I was a bit shy. I think I would’ve been more relaxed if we had met face-to-face, as we could’ve broken the ice. However, this experience allowed me to practice my interview skills when I talked with the women”.

In the case mentioned above, the students were expected to conduct informal interviews with members of the public who were part of the target audience to inform the development of the social marketing plan, which was the main output of the course.

### 5.4. The Impact of The Course beyond The Classroom

Some community-partner organizations implemented the social marketing plans developed during the course, regardless of the delivery mode. For example, the plan created by the students enrolled in the traditional section in Spring 2018 was turned into a research proposal funded by the Global Compact Network Lebanon (GCNL). The project aimed to promote colorectal-cancer screening among vulnerable individuals in Lebanon, and it was conducted in collaboration with SAID NGO and colleagues from the School of Nursing. As for Fall 2019, the project addressing organic-waste composting was carried over in Spring 2021 in an e-service learning undergraduate course on social marketing (HPCH 204) taught entirely online by the first author. Two social marketing plans developed in Fall 2020 were carried over in 2021—and one of them resulted in a recent publication [70]. Another community-partner organization turned the mental-health-support strategy into a research proposal, which MasterCard Foundation funded; the strategy will be implemented in the upcoming academic year 2021–2022. The Humanitarian Engineering Initiative is currently finalizing a platform addressing COVID-19 misinformation (project “*Wikaytek*”), which was implemented in the Fall 2021 semester.

## 6. Discussion

In this case study, we presented our experience introducing e-service learning in a social marketing course that was traditionally taught face to face before recent events forced the instructors to redesign the course to be delivered entirely online. We examined the impact of e-service learning over time, considering indicators such as the student’s engagement in the course, the attainment of the learning outcomes, satisfaction with the course, and impact beyond the classroom.

### 6.1. Impact of E-Service Learning on Student Learning

Compared with the international evidence on e-service learning, this pedagogy is a powerful strategy that creates meaningful learning experiences for students and community partners. Despite structural limitations, such as the access to Internet resources common in low-resource countries or settings such as Lebanon, our e-service learning course did improve the learning of social marketing or the knowledge transfer between the students and communities [41,46]. We found that, if carefully designed and appropriately implemented, e-service learning can improve the experience with and learning of social marketing, which is consistent with the literature on service learning in the public health education [2,5,6].

The quantitative evaluations and feedback collected at the end of each semester suggested that all the students enjoyed the course and rated the course very highly. Regarding student engagement, e-service learning allows students to learn flexibly (online), giving them more time to engage in service activities, which is aligned with previous reports [10,34]. The flexibility of the approach was more evident in the latest iterations of the course when the students were required to perform all the activities remotely. The students perceived that the e-service learning course improved their critical decision-making and time-management skills, as well as their interpersonal communication skills, which aligns with previously reported studies and reviews [45,46,47,49]. According to the final ICE reports, the students participating in e-service learning felt they learned much through the course. Using a different participation strategy, the instructor set expectations upfront, motivating most of the students and maintaining effective involvement and engagement in the class. The role of the instructor in boosting student engagement has been previously reported in the service learning literature [71] and specifically in the e-service learning literature [45,46,47]. Our findings are consistent with two applications of blended learning in public health technical courses in survey design and biostatistics, in which the authors observed an improved student performance in the blended learning sections [7,8]. Our findings also confirm the e-service learning literature [54] and match what is reported in the social marketing literature concerning using active learning pedagogies [32].

E-service learning helps teach social marketing, allowing students to flexibly reach out to the served communities and promote change within the classroom [33]. E-service learning can provide an ideal environment for social marketers interested in building community-academic partnerships, as reported in other disciplinary case studies [10,34].

### 6.2. The Impact of E-Service Learning beyond the Classroom

Even though the students faced many contextual and environmental challenges in 2019–2020, many perceived e-service learning as an enriching yet demanding experience that would benefit their careers and the community they served; this finding aligns with the high demands of service learning courses that instructors, students, and community partners perceive that were reported in a compilation of case studies from Europe and Lebanon [10].

The qualitative feedback collected through the end-of-year ICE reports and written reflections shows that e-service learning was considered “heavy” and effortful, consistent with the literature on service learning courses [72]. The pace of e-service learning courses is rigorous and can be challenging. Learners are expected to apply concepts in real-life settings, all while developing skills and acquiring new knowledge [2,4]. An effective strategy to reduce the perceived workload was the Karma Participation point system introduced in Fall 2020. This nontraditional assessment method focused on the learning process rather than on standard evaluative exams and research papers [5]. In our study, the students found value in the Karma Points as an interactive method that created a supportive environment for learning through participation and engagement with peers and instructors.

Constrained by the COVID-19 lockdowns, our students struggled with working on projects without the possibility of meeting in person with their peers and community partners. This limitation has also been reported in other recent examples of e-service learning courses delivered during the pandemic [41,42,43,44]. Although e-service learning can enhance soft skills among students, one of the biggest challenges is time management. While e-service learning can offer a more flexible learning environment for public health students, instructors need to set clear expectations and demonstrate the benefits of online vs. face-to-face learning, showing students how to use software and how to use their time online offline. This can be achieved by breaking complex project outcomes into smaller achievable milestones that can be reviewed and assessed throughout the course. Students come with various prior experiences in teambuilding, communication, the division of tasks, and coordination, all of which are learning opportunities and keys to succeeding in e-service learning activities [45,46,49]. Instructors should be mindful of these differences when setting up the course plan. They should also provide learners with tools and activities to probe their progress and maintain healthy communication with community partners.

Because service learning instills civic engagement and social responsibility among students, social marketing or other disciplines can benefit from this pedagogy by implementing programs that solve real public health problems [33]. This case study confirms what has been reported in a compilation of case studies on social marketing courses offered in Europe and Lebanon (by the first author) in 2019 [10]. However, the impact of e-service learning beyond the classroom can be found in a more recent example, in which the data collected in a recent study by the first author [70] were used to inform a social marketing plan developed by students participating in the same social marketing course.

In the future, even if we might not go back to how it was before COVID-19, the social marketing course will still contain an e-service learning component for the instructional part, as we have invested time and resources in developing self-paced Moodle lessons that can be reused and upscaled. These resources will allow students and instructors to spend more time interacting with community partners and having active discussions in class.

### 6.3. Implications for Education and Practice

This case study bears some considerations for colleagues teaching social marketing or similar disciplines in institutions facing restrictions on face-to-face instruction. E-service learning can efficiently achieve learning outcomes, and it impacts student engagement and participation in the course.

Instructors who want to bring innovation to their social marketing courses should try e-service learning; they should invest time in designing blended or fully online courses through self-paced lessons and interactive activities to maintain the students’ engagement and contact with community partners. Blended learning offers a more flexible learning environment for public health students enrolled in service learning courses; instructors need to set clear expectations [71] and demonstrate the benefits of online vs. face-to-face learning, showing the students how to use software and how to make good use of their time online and offline. Finally, e-service learning offers an excellent opportunity to engage with local communities when face-to-face interactions are limited or unfeasible. Some community partners might be reluctant to engage online. Hence, instructors must plan such interactions carefully, trying not to impose but to accommodate the community partners’ needs and communication habits. It is essential to demonstrate how communication and information sharing happen using online platforms to gain support.

At the same time, instructors should pay attention to the Internet infrastructure and digital literacy of both students and community partners to minimize the potential divide that the introduction of new digital tools might create, thereby exacerbating inequalities among students or communities.

### 6.4. Limitations

This case study has some limitations, including the relatively small number of students involved and the lack of an experimental study design. The social marketing course is offered to all MPH students specializing in health promotion, community health, and public health nutrition, with cohorts averaging 20 students per year. This paper presents all the feedback received, including the students’ views who voluntarily expressed their opinions through anonymous comments. Another factor to consider is the highly challenging context in which the students had to work and operate, with overarching challenges that might have aggravated underlying inequalities in terms of Internet connectivity and digital literacy. Regarding the experimental design, in the context of educational studies, experimental studies are not only hard to implement due to university policies but also inequitable. For example, in 2018, the blended section was almost double the size of the traditional one because more students wanted to enroll in one of the two sections. While this choice was based on convenience and feasibility (better-fitting schedules), we cannot rule out a potential selection bias.

## 7. Conclusions

Based on our experience with e-service learning for teaching a social marketing course, we recommend this approach to any instructor interested in delivering similar courses and developing solutions that impact students and local communities. E-service learning allowed us to adapt to situations where face-to-face contact could not be maintained due to external factors. This allowed us to manage the time better and respond to uncertainties by offering continuous support to students while meeting the learning objectives. To ensure the optimal delivery of e-service learning courses, instructors should ensure that students have the technical skills and capabilities to utilize the modes of delivery. If not, the instructors should allocate some time to provide guidance and offer social and technical support to the students who need that.

E-service learning can be instrumental in other contexts and settings hit by crises or emergencies. Instructors are essential in coordinating and guiding e-service learning courses; they must invest time and resources to organize the course and set protocols and channels to communicate with the students and “client” organizations. Instructors should also consider using nontraditional participation- and engagement-assessment methods in e-service learning courses that encourage process-based learning.

E-service learning also allows for continuity in the relationship with the community-partner organizations when physical presence and engagement are not feasible. Technology can enable and catalyze partnerships between academia and community organizations.

## Figures and Tables

**Figure 1 ijerph-19-12696-f001:**
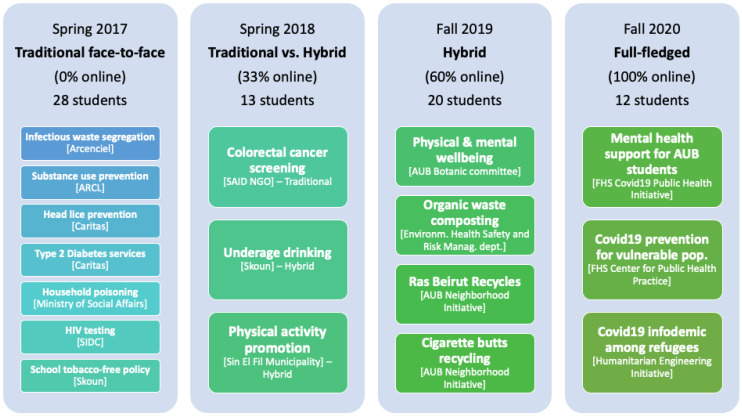
Characteristics of four iterations of the course and topics addressed.

**Table 1 ijerph-19-12696-t001:** Impact indicators of e-service learning course.

Indicators	Spring 2017 (0% Online) *(n* = 28)	Spring 2018-1 (0% Online) (*n* = 5)	Spring 2018-2 (33% Online) (*n* = 8)	Fall 2019 (60% Online) (*n* = 20)	Fall 2020 (100% Online) (*n* = 12)
** *Engagement in the course* **					
Participation grade (%) (SD)	93 (8.6)	90 (10.0)	93 (7.1)	80 (25.1)	83 (12.6)
** *Learning outcomes* **					
Course learning outcomes (5-point scale)	3.8	4.3	4.4	4.4	4.8
Final-project grade (%) (SD)	91 (3.7)	86 (0.0)	82 (1.3)	97 (2.7)	98 (1.5)
** *Satisfaction with…* **					
Course (5-point scale)	4.2	4.1	4.5	4.4	4.6
Instructor (5-point scale)	4.1	4.3	4.5	4.5	4.9
Delivery mode (5-point scale)	4.1	4.3	4.5	4.5	4.8

Notes: a post hoc comparison with the traditional section was significant at *p* < 0.001 for Spring 2018, Fall 2019, and Fall 2020.

## Data Availability

The datasets used are available from the corresponding author upon request.

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
