# Peer review of "Teaching Social Marketing Using E-Service Learning Amidst Health and Humanitarian Crises: A Case Study from Lebanon"

_ijerph, 2022, doi:10.3390/ijerph191912696_

Round 1

Reviewer 1 Report

Dear Authors,

Thank you for submitting this very interesting paper, which I overall liked a lot. I provide below some comments and minor revision recommendations on how to improve the paper, in terms of the literature review and contribution.

The paper can be further developed in the section around social marketing which seems slightly underdeveloped. I would suggest the authors to include a couple of paragraphs which elaborate on the growth and evolution of social marketing literatures and approaches, especially since the field is growing exponentially the last decades, to tackle societal challenges. I provide below use useful papers which the authors might wish to cite.

Hastings and Saren (2003) The Critical Contribution of Social Marketing: Theory and Application Marketing Theory, vol. 3, 3: pp. 305-322. 

Duane, S., and Domega, C., (2018) Social marketing partnerships: Evolution, scope and substance Marketing Theory, vol. 19, 2: pp. 169-193. , 

Peattie and Peattie (2003) Ready to Fly Solo? Reducing Social Marketing’s Dependence on Commercial Marketing TheoryMarketing Theory, vol. 3, 3: pp. 365-385.

In the section Literature Review, I will invite the authors to adopt a more culture sensitive perspective and avoid generalizations. Depending on Internet Connectivity, affluence, government infrastructures, online provision varies from country to country and even from neighbourhood to neighbourhood in megacities. You provide only one sentence about the ‘South World’ but you could include more examples regarding the differences in several Western contexts (apart from Australia and Alabama) and how you perceive the Global South. Maybe a definition? I would recommend to expand with a paragraph.

You methods and methodology are sound. I am aware – and I might be totally wrong – that Lebanon was facing economic and social turmoil before 2019 and I am a bit surprised with World Bank’s classification, so perhaps you could elaborate here on the Lebanese context here?

In the findings section, try to connect your data and findings with social marketing literatures. I am afraid the connection is loose here and that’s one of the reasons I asked you to expand on the literature review. How the intersection of e-learning and social marketing can be adopted applied by other academic and practitioners?

Again, this is a well-structured paper.

Author Response

Response to Reviewer 1 Comments

Dear Authors,

Thank you for submitting this very interesting paper, which I overall liked a lot. I provide below some comments and minor revision recommendations on how to improve the paper, in terms of the literature review and contribution.

Response 1.1. Thank you for your positive and constructive comments. We have amended the manuscript to address your and the other reviewers’ suggestions to the best of our ability. Please find below our point-by-point response.

The paper can be further developed in the section around social marketing which seems slightly underdeveloped. I would suggest the authors to include a couple of paragraphs which elaborate on the growth and evolution of social marketing literatures and approaches, especially since the field is growing exponentially the last decades, to tackle societal challenges. I provide below use useful papers which the authors might wish to cite.

Hastings and Saren (2003) The Critical Contribution of Social Marketing: Theory and Application Marketing Theory, vol. 3, 3: pp. 305-322. 

Duane, S., and Domega, C., (2018) Social marketing partnerships: Evolution, scope and substance Marketing Theory, vol. 19, 2: pp. 169-193. 

Peattie and Peattie (2003) Ready to Fly Solo? Reducing Social Marketing’s Dependence on Commercial Marketing TheoryMarketing Theory, vol. 3, 3: pp. 365-385.

Response 1.2. Thank you for the suggestion to expand the paragraph on social marketing. Since even the other reviewer made a similar comment, we decided to expand the paragraph on social marketing, including some literature on its scholarship and pedagogy. As members of the International and European Social Marketing Associations, we are aware of the developments in this field of study. We have included some more recent literature to support our statements, but we appreciate the suggested references.

In the section Literature Review, I will invite the authors to adopt a more culture sensitive perspective and avoid generalizations. Depending on Internet Connectivity, affluence, government infrastructures, online provision varies from country to country and even from neighbourhood to neighbourhood in megacities. You provide only one sentence about the ‘South World’ but you could include more examples regarding the differences in several Western contexts (apart from Australia and Alabama) and how you perceive the Global South. Maybe a definition? I would recommend to expand with a paragraph.

Response 1.3. We have revised the literature review, adding a paragraph defining the concept of Global South.

You methods and methodology are sound. I am aware – and I might be totally wrong – that Lebanon was facing economic and social turmoil before 2019 and I am a bit surprised with World Bank’s classification, so perhaps you could elaborate here on the Lebanese context here?

Response 1.4. As we mentioned above, we elaborated on Lebanese context in the introduction as follows…

Lebanon certainly belongs to this group of spaces, especially after the most recent social, political, economic, and financial crises [57,58], which still affect the small country in the Eastern Mediterranean region. Emerged in October 2019 [58], a fiscal and financial crisis revealed deep-rooted socio-political issues. The COVID-19 pandemic, whose outbreak started in February 2020, and the Beirut Blast in August 2020 [59], shook the pillars of an already weakened healthcare system, exacerbating pre-existing, underlying social, and political inequities. These circumstances made it difficult to prioritize food, health, education, and other essential services such as water, electricity, and the internet, which has recently become a commodity. In Lebanon, the national power grid is so limited and dysfunctional that a parallel, decentralized system provides electricity to households and institutions through diesel-based generators [60]. Prone to corruption and monopoly, the availability of electricity and the internet depends on the supply and demand of constantly increasing fuel costs. On some occasions, even the internet providers did not have the fuel to power their generators, resulting in significant service disruptions, especially in rural areas outside the capital Beirut. This has likely affected the students’ learning experience and made it difficult for us to implement online-based education, even though it was pushed by the University.

In the findings section, try to connect your data and findings with social marketing literatures. I am afraid the connection is loose here and that’s one of the reasons I asked you to expand on the literature review. How the intersection of e-learning and social marketing can be adopted applied by other academic and practitioners?

Response 1.5. We have revised the findings section based on your comment and provided additional commentary on social marketing cases.

Again, this is a well-structured paper.

Response 1.6. Thank you so much for your comments.

Reviewer 2 Report

Thank you for a timely and interesting paper - congratulations on the research done.

Front end:

Perhaps 'competency' is not the right word to describe social marketing (page 2), primarily when you use the word 'discipline' in the following sentence.

The discussion on social marketing in paragraph 2 on page 2 needs revisiting. The theoretical foundation of social marketing requires expansion and elaboration. Don't assume that all readers are familiar with social marketing and its application in Public Health. Social marketing is more just a 'competency'; the field is well established in the behavior change arena, and the discussion here must reflect on this.

Additionally, it will be worth looking into social marketing pedagogy as the aim of this study links with e-learning services.

Regarding the study's reasoning, please provide further justification for your choices, including why the application of service learning in social marketing education is limited.

Literature review

There is very little literature review prior to embarking on methods.

The discussion on blended learning on page 3 ends abruptly. Do offer more critique to explain your point. Similarly, the two examples offered in the last paragraph of the literature review on page 4 require further criticality and a strong link to the aim of your study. Unpacking these examples would add more weight to the analysis presented.

The point on the western voice and lack of the global south stance is very important and requires further analysis.

Discussion

The discussion highlights some very interesting points; however, a more thorough/critical approach would be useful for the audience of this journal. I would recommend a critical discussion on the contribution of this study (implications for social marketing pedagogy or those who teach social marketing programs/courses/modules) and its limitations.

Would there be any implication for developing community-based social marketing courses? This can be linked with Community Based Social Marketing by Doug McKenzie-Mohr, a well-established approach in social marketing. This can then be linked with the wider impact of such approaches, e.g., the impact on students' careers, community partners, educational institutes, and bringing all parties together (i.e., impact beyond teaching and learning)

A reference to the lack of a non-western voice is made earlier in the literature review, discussion on how this study would overcome such a gap would be useful.

Similarly, a reflection on the study's sample size and the negative feedback, if received from students (or constructive comments, for that matter), would add more value to the argument presented.

Overall comments

·       A more robust approach must be adopted to discuss social marketing, especially in relation to the context of this study

·       A more critical approach must be used to review the relevant literature

·       There are minor typing and edits to be made, e.g., on lines 55-56, even though 55 social marketing remains taught chiefly in business and communications schools [12–14]; I think you might like to re-write this sentence for more clarity.

·       Do a full and detailed ref, spelling, and typing check of the paper.

This is a promising study. I hope my comments will prove helpful to you. Good luck!

Author Response

Response to Reviewer 2 Comments

Thank you for a timely and interesting paper - congratulations on the research done.

Response 2.1. Thank you for your positive comments. We have amended the manuscript to address the points you raised below. Please find below our point-by-point response.

Front end:

Perhaps 'competency' is not the right word to describe social marketing (page 2), primarily when you use the word 'discipline' in the following sentence. 

Response 2.2. We used the word competency as defined by the Association of Schools of Public Health in the European Region (ASPHER), which listed social marketing under the “intellectual competencies”. We are well aware that social marketing is more of a discipline or a field of study. However, we revised the sentence to keep the word in quotation marks to reflect the ASPHER definition.

The discussion on social marketing in paragraph 2 on page 2 needs revisiting. The theoretical foundation of social marketing requires expansion and elaboration. Don't assume that all readers are familiar with social marketing and its application in Public Health. Social marketing is more just a 'competency'; the field is well established in the behavior change arena, and the discussion here must reflect on this. 

Additionally, it will be worth looking into social marketing pedagogy as the aim of this study links with e-learning services.

Response 2.3. Point well taken. We have revised and restructured the introduction and literature section to dedicate more space to the definition of social marketing scholarship and education/pedagogy.

Regarding the study's reasoning, please provide further justification for your choices, including why the application of service learning in social marketing education is limited.

Literature review

There is very little literature review prior to embarking on methods. 

Response 2.4. We have revised the literature section to expand on service learning applications and social marketing, including some known examples. To the best of our knowledge, there is no published evidence on the use of service learning in social marketing beyond a previously published book chapter by the first author (Bardus et al. 2019) and a conceptual paper by Domegan and Bringle (2010).

The discussion on blended learning on page 3 ends abruptly. Do offer more critique to explain your point. Similarly, the two examples offered in the last paragraph of the literature review on page 4 require further criticality and a strong link to the aim of your study. Unpacking these examples would add more weight to the analysis presented. 

Response 2.5. We have revised the discussion to provide more elaboration to the examples included, knowing that the evidence is nearly absent.

The point on the western voice and lack of the global south stance is very important and requires further analysis. 

Response 2.6. Thank you for pointing this out. Based on the other reviewer’s comment, we have expanded the paragraph regarding the voices from the Global South as an important element in the epistemological diversity debate, recently discussed in a paper by Cateriano et al. (2022).

The discussion highlights some very interesting points; however, a more thorough/critical approach would be useful for the audience of this journal. I would recommend a critical discussion on the contribution of this study (implications for social marketing pedagogy or those who teach social marketing programs/courses/modules) and its limitations. 

Response 2.7. Thank you for this suggestion. We added paragraph 6.3 to address the implications for practice and paragraph 6.4 for the limitations.

Would there be any implication for developing community-based social marketing courses? This can be linked with Community Based Social Marketing by Doug McKenzie-Mohr, a well-established approach in social marketing. This can then be linked with the wider impact of such approaches, e.g., the impact on students' careers, community partners, educational institutes, and bringing all parties together (i.e., impact beyond teaching and learning)

Response 2.8. Thank you for this suggestion. We added a recently published example of how a research project was used as a basis for developing a social marketing plan throughout the course (reference #70).

A reference to the lack of a non-western voice is made earlier in the literature review, discussion on how this study would overcome such a gap would be useful. 

Response 2.9. While it is an important aspect, we believe our case study does not allow us to overcome such a gap.

Similarly, a reflection on the study's sample size and the negative feedback, if received from students (or constructive comments, for that matter), would add more value to the argument presented. 

Response 2.10. We elaborated on the sample size and feedback on the case study's limitations (paragraph 6.4).

Overall comments

  • A more robust approach must be adopted to discuss social marketing, especially in relation to the context of this study 
  • A more critical approach must be used to review the relevant literature

Response 2.11. We have rewritten the introduction and the literature review paragraph, including a paragraph on social marketing, providing a critical review of the literature.

  • There are minor typing and edits to be made, e.g., on lines 55-56, even though 55 social marketing remains taught chiefly in business and communications schools [12–14]; I think you might like to re-write this sentence for more clarity. 
  • Do a full and detailed ref, spelling, and typing check of the paper.

 Response 2.12. We have run a spell-check using two different software (Word language editor and Grammarly), and we have also enlisted a native speaker to check typos and edits.

This is a promising study. I hope my comments will prove helpful to you. Good luck!

Response 2.13. Thank you, your comments were very helpful, and we hope that the manuscript is improved to an acceptable level.